# InfinityStar: Unified Spacetime AutoRegressive Modeling for Visual Generation

**Jinlai Liu***, **Jian Han***, **Bin Yan***  **Hui Wu,**  **Fengda Zhu,**  **Xing Wang**

**Yi Jiang,**  **Bingyue Peng,**  **Zehuan Yuan**[†]

ByteDance

{liujinlai.licio,hanjian.thu123,bin.yan,wuhui.321,fengdazhu}@bytedance.com,
{xing.wang,jiangyi.enjoy,bingyue.peng,yuanzehuan}@bytedance.com,

Codes and models: https://github.com/FoundationVision/InfinityStar

## Abstract

We introduce InfinityStar, a unified spacetime autoregressive framework for high-resolution image and dynamic video synthesis. Building on the recent success of autoregressive modeling in both vision and language, our purely discrete approach jointly captures spatial and temporal dependencies within a single architecture. This unified design naturally supports a variety of generation tasks such as text-to-image, text-to-video, image-to-video, and long-duration video synthesis via straightforward temporal autoregression. Through extensive experiments, InfinityStar scores 83.74 on VBench, outperforming all autoregressive models by large margins, even surpassing diffusion competitors like HunyuanVideo. Without extra optimizations, our model generates a 5s, 720p video approximately $10\times$ faster than leading diffusion-based methods. To our knowledge, InfinityStar is the first discrete autoregressive video generator capable of producing industrial-level 720p videos. We release all code and models to foster further research in efficient, high-quality video generation.

## 1 Introduction

Visual synthesis has witnessed remarkable progress in recent years, largely propelled by the scaling of Transformer architectures. In particular, video generation has attracted growing interest from both academia and industry, owing to its wide-ranging applications in content creation, world simulation, etc. At present, diffusion models[3, 17, 16, 26, 7, 37] lead the field by iteratively denoising latent representations to produce high-fidelity clips. Concurrently, autoregressive models[15, 28, 8] have been explored for their potential to unify image and video generation and to generalize over longer time horizons.

Despite their successes, each paradigm exhibits critical shortcomings. Video diffusion models excel at synthesizing fixed-length frame sequences by exploiting bidirectional attention, yet they incur substantial computational cost due to tens or even hundreds of sequential denoising steps, and they struggle to extend seamlessly to video extrapolation. Autoregressive methods based on next-token prediction, while inherently capable of streaming generation, often fall short in visual fidelity and suffer from prohibitive latency due to tens of thousands of inference steps.

These observations motivate the need for a generation framework that simultaneously possess high visual quality, efficiency and temporal generalization. Recently, Visual AutoRegressive modeling (VAR)[23] redefined image generation as a coarse-to-fine next-scale prediction. Its follow-up work, Infinity [12] further introduces bitwise modeling and scales up the vocabulary size, achieving

---

*Equal contribution. [†]Corresponding author: yuanzehuan@bytedance.com

39th Conference on Neural Information Processing Systems (NeurIPS 2025).

comparable performance to diffusion models while offering significant advantages in inference speed. Inspired by the success of VAR [23] and Infinity [12], we present InfinityStar, a Spacetime Pyramid Modeling for unified text-to-image, text-to-video, zero-shot image-to-video, and zero-shot video extrapolation. This framework models a video as an image pyramid and multiple clip pyramids, not only naturally inheriting the text-to-image capabilities but also decoupling static appearance from dynamic motions in videos. Furthermore, we introduce several key improvements. First, we improve discrete reconstruction quality by leveraging knowledge inheritance from a continuous video tokenizer. Second, we introduce Stochastic Quantizer Depth during training of the tokenizer to alleviate the imbalanced information distribution across scales. Third, we propose Semantic Scales Repetition, which refines the predictions of earlier semantic scales in a video, significantly enhancing fine-grained details and complex motions of the generated videos.

We train InfinityStar on large-scale video corpora to support up to 720p resolution and variable durations. On the VBench benchmark[36], InfinityStar establishes a new state-of-the-art among autoregressive video models, even surpassing industry-leading HunyuanVideo[16] (83.74 v.s 83.24). Besides, InfinityStar shows a great advantage in terms of speed. Using visual tokenizers of the same compression rate, InfinityStar achieves a $10\times$ reduction in inference latency relative to leading diffusion models.

In summary, the main contributions of our work are as follows:

1. We propose InfinityStar, a novel spacetime pyramid modeling framework that unifies diverse visual generation tasks, demonstrating superior flexibility and versatility.

2. InfinityStar is the first discrete autoregressive model capable of generating high-quality videos, outperforming existing autoregressive text-to-video models and matching the performance of leading diffusion models.

3. Compared to the inefficiency of existing autoregressive models and diffusion models, InfinityStar significantly accelerates high-quality video generation.

## 2 Related Work

### 2.1 Video Diffusion Models

Diffusion models excel at generating high-fidelity data by gradually denoising random noise and can naturally extend to video generation. Early attempts [2, 4, 34] are built on U-Net architectures, demonstrating the feasibility of this approach but falling short in producing sharp, temporally coherent frames due to limited model capacity. The advent of Diffusion Transformers (DiT [19]) marked a turning point. SORA [3] harnessed DiT's scaling ability to process spatio-temporal patches at scale, dramatically improving both video consistency and generation quality. Inspired by SORA's success, industry efforts [32, 16, 26] have further advanced the field, pushing video generation to new heights. Although video diffusion models deliver outstanding quality, their slow generation speed hinders the production of high-resolution, long-duration videos.

### 2.2 Video AutoRegressive Models

Another class of methods [28, 8, 15] employs autoregressive models for video generation. Inspired by the success of LLMs, these works predict video tokens in specific orders using an autoregressive Transformer. For example, Emu3 [28] performs next-token prediction along both spatial and temporal axes, while NOVA [8] first predicts spatial tokens set-by-set and subsequently proceeds frame-by-frame in the temporal dimension. Although achieving preliminary progress, they require hundreds to thousands of inference steps, resulting in prohibitively low generation efficiency. In contrast, recent advances in next-scale prediction [23, 12] have demonstrated state-of-the-art performance in image synthesis, offering both improved quality and markedly faster inference. In this work, we extend the next-scale prediction paradigm to the unified tasks of image and video generation.

### 2.3 Discrete Video Tokenizers

For a long time, discrete [31, 33] and continuous [16, 32, 26] video tokenizers have been developed independently. Although some works [1, 27] provide both discrete and continuous tokenizers, the

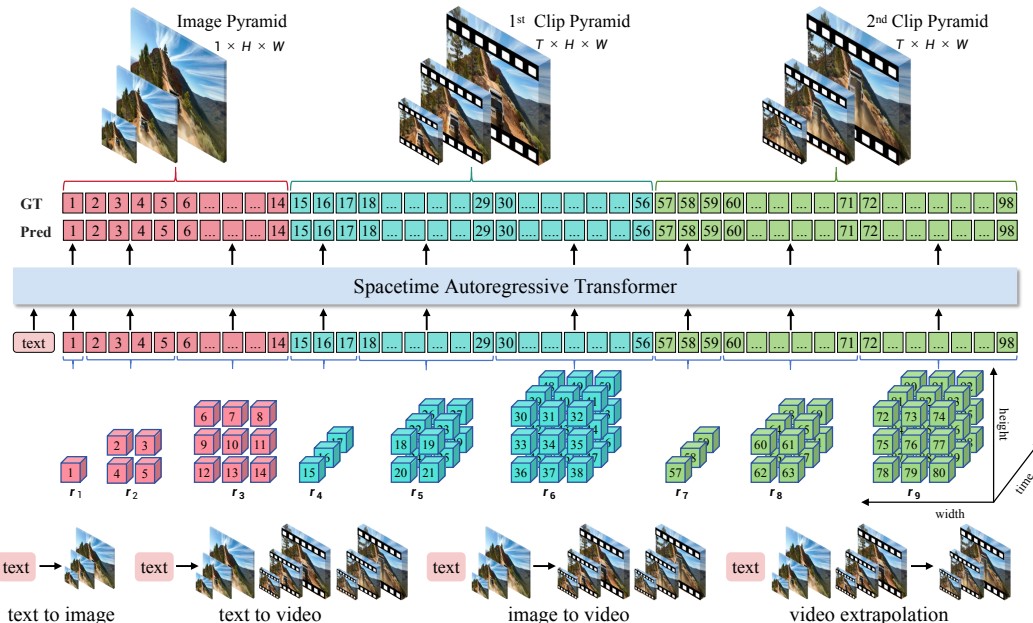

Figure 1: **Spacetime pyramid modeling of InfinityStar.** Built with an unified autoregressive pipeline, InfinityStar is capable of performing text-to-image, text-to-video, image-to-video, video extrapolation tasks all in one model.

network configurations are usually not aligned. For example, Cosmos [1] chooses 6 and 16 as latent dimensions in its discrete and continuous variants respectively. This misalignment hinders the knowledge reuse between two types of tokenizers. As a result, most mainstream discrete video tokenizers are either trained from scratch [1] or starting from a pretrained discrete image tokenizer [33, 27]. However, these training strategies have the following drawbacks. First, training from scratch is inefficient and converges slowly. Second, weights pretrained on static images are not optimal for video reconstruction. To alleviate these deficiencies, we propose a new training strategy, which inherits the architecture and knowledge of a trained continuous video tokenizer. Experiments show that this strategy significantly boosts the convergence of discrete video tokenizers.

# 3 InfinityStar Architecture

## 3.1 Preliminaries

**Infinity for Image Generation.** Infinity [12] decomposes an image into a sequence of hierarchical token blocks using a visual tokenizer and models the relationship between tokens by a visual autoregressive Transformer (VAR Transformer). To cover images of various sizes, Infinity pre-defines a list of token block sizes $\{(h_1, w_1), ...(h_K, w_K)\}$, called scale schedule. The size $(h_i, w_i)$ in scale schedule grows as $i$ increases, forming a pyramid-like structure, which we refer as **image pyramid** in later discussion. Next we introduce the training and inference procedure of Infinity.

In the first training stage, a visual tokenizer learns to reconstruct the raw image and compress it into a sequence of discrete tokens, which can be modeled by the VAR Transformer in the next stage. Specifically, the tokenizer first encodes the raw images into compact latents, then transforms latents into $K$ discrete residual token blocks $(r_1, r_2, ..., r_K)$ using a bitwise multi-scale residual quantizer [12]. Each token block $r_i$ consists of $h_i \times w_i$ discrete tokens of $d$-dim with vocabulary size of $2^d$. Then in the second stage, a VAR Transformer is trained to predict next residual token block $r_k$ conditioned on text embedding $\psi(t)$ and former tokens blocks $r_{<k}$. Formally, in each step, VAR Transformer predicts a conditional probability $p(r_k|r_{<k}, \psi(t))$. During the inference, Infinity generates an image by running the VAR Transformer $K$ times autoregressively, merging the predicted tokens and running the tokenizer decoder once.

## 3.2 Spacetime Pyramid Modeling for Unified Generation

Extending the spatial-only next-scale prediction paradigm of Infinity [12] to video generation presents a primary challenge: *how to incorporate the temporal dimension*. The straightforward strategies are either letting time grows uniformly, *i.e.*, from $(1, 1, 1)$ to $(T, H, W)$, or keeping time constant, *i.e.*, from $(T, 1, 1)$ to $(T, H, W)$. We empirically found that letting time grow uniformly produces flickering videos. As for the constant time pyramid, we refer to it as the **pseudo-spacetime pyramid**. Despite its conceptual simplicity, it suffers from two fundamental limitations. First, the treatment of videos differs markedly from that of images, preventing a text-to-video (T2V) model from effectively leveraging the knowledge learned by a text-to-image (T2I) model and complicating its extension to tasks such as image-to-video (I2V). Second, because appearance and motion in videos are coupled in this design, the model faces significant difficulty in accurately fitting both aspects.

To overcome these challenges, we propose a novel **spacetime pyramid modeling** framework as shown in Fig.1. Each video is decomposed into sequential clips $\{c_1, c_2, \cdots, c_N\}$. We regard the first frame as $c_1$ (*i.e.*, $T = 1$) to encode video main static appearance cues specifically and other clips share an equal duration $T > 1$. Each clip is modeled as a 3D volume pyramid similar as Infinity [12]. In particular, for each clip, there are $K$ scales with each represented as a residual token block $r_k$ of $(T, h_k, w_k)$ dimension. *It is worth noting that all scales in the pyramid are extended only in spatial dimension instead of time*. Mathematically, the clip tokens are generated auto-regressively across scales as:

$$p(r_1^1, \ldots, r_K^1) = \prod_{k=1}^{K} p(r_k^1 \mid r_1^1, \ldots, r_{k-1}^1, \psi(t)), \tag{1}$$

For inter-clip predictions, clips are generated sequentially conditioned on prior clip predictions and the text input in an autoregressive manner. In this way, we could generate infinitely long videos theoretically. Formally, the autoregressive likelihood of the whole video can be expressed as:

$$p(r_1^1, \ldots, r_K^N) = \prod_{c=1}^{N} \prod_{k=1}^{K} p(r_k^c \mid r_1^1, \ldots, r_{k-1}^c, \psi(t)), \tag{2}$$

## 3.3 Visual Tokenizer

Training video tokenizers faces greater challenges than training image tokenizers. First, training tokenizers on videos of tens of frames is much computationally heavier than training on static images. Therefore, training a video tokenizer from scratch is extremely time-consuming and suffers from slow convergence. Second, the scale schedule in videos leads to more imbalanced information distribution, where most information is concentrated in the last few scales. This brings great difficulties to the optimization of VAR Transformer. To solve these challenges, we introduce two techniques, knowledge inheritance from continuous video tokenizer and stochastic quantizer depth.

**Knowledge Inheritance from Continuous Video Tokenizer**. Instead of designing and training a discrete video tokenizer from scratch, we inherit the architecture and weights of a trained continuous video tokenizer, *i.e.* video VAE. Specifically, we first insert a parameter-free quantizer between the pre-trained VAE encoder and the decoder. The quantizer is based on binary spherical quantization [38], being similar to that of Infinity [12] but with new spacetime pyramid scale schedule. This does not introduce any new parameter like codebook in VQ [24] and well retains knowledge of the original VAE. As shown in Fig.5, the discrete video tokenizer reconstructs videos decently, even without any fine-tuning. To further improve the reconstruction quality, we fine-tune the new tokenizer jointly on images and videos like previous works [27, 1]. During the fine-tuning, the KL loss of the original VAE is replaced with the commitment loss plus the entropy penalty [38]. As shown in Fig.5, with the help of knowledge of continuous video VAE, the convergence accelerates dramatically.

**Stochastic Quantizer Depth**. When tokenizing videos using the spacetime pyramid schedule, the information distribution on different scales gets extremely imbalanced. Specifically, there are only a few tokens in the early scales, while there are tens of thousands of tokens in the last scales. Thus the tokenizer tends to reconstruct videos solely relying on tokens from the last few scales and not to learn useful representation in early scales as shown in Fig.6 (left). However, this imbalanced distribution is difficult to model using VAR Transformer because the dependence between the latter token blocks and the former ones is weak. To alleviate this problem, we propose a regularization called stochastic quantizer depth. During training, each one of the last $N$ scales has a probability $p$ of being discarded.

In this way, there are $2^N$ possible scale schedules during training. This requires the tokenizer to reduce the reliance on last scales and store more information in tokens of early scales. As in Fig.6 (right), with the help of this regularization, the reconstruction results of early scales become much clearer. This balanced information distribution makes the training of VAR Transformer easier.

### 3.4 Spacetime Autoregressive Transformer

To accommodate the newly introduced temporal dimension, enhance the quality of generated videos, and alleviate the substantial computational overhead associated with a large number of tokens, we propose the following modifications to the VAR Transformer: Spacetime RoPE, Semantic Scale Repetition, and Spacetime Sparse Attention. We put Spacetime RoPE in the appendix.

**Semantic Scale Repetition.** With carefully crafted positional encodings, InfinityStar can already generate videos of acceptable quality. However, we observe that the structural coherence and motion dynamics in these outputs remain suboptimal. As shown in Figure 6, the overall layout and the placement of foreground objects are determined by the early scales of the clip pyramid—what we term the "semantic scales." This observation motivates us to enhance generation fidelity at these semantic scales. To this end, we introduce a simple yet effective technique called semantic scale repetition. Concretely, if a clip pyramid comprises $K$ scale tuples, we repeat the first $K_s$ tuples $N$ times, thereby reinforcing the semantic-level representations. In this way, every earlier residual $r_k$ undergoes multiple rounds of refinement, improving the generation quality of semantics and the performance in complex scenarios with large motion. Given that the tokens at these early scales account for only a small fraction of the total token count, the additional computational overhead incurred by repeating them is negligible.

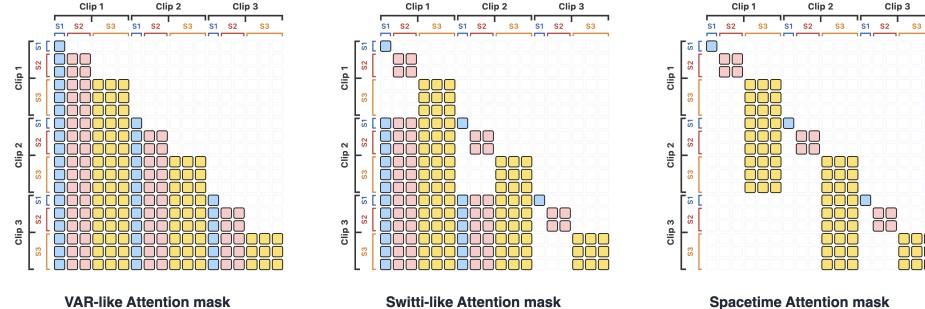

**Figure 2:** Illustration of three causal attention variants. We plot three pyramids on the scale size = (1,2,3) for visualization simplicity. From left to right, VAR block-wise causal mask with full history, Switti block-wise non-causal mask with full history, and spacetime sparse attention.

**Spacetime Sparse Attention.** Autoregressive video generation faces significant challenges due to the high computational costs of long context. As on the left of Fig.2, Infinity [12] employs a block-wise causal mask for single pyramid modeling. Switti [25] verifies that conditioning next-scale predictions solely on inputs from preceding scales is sufficient, resulting in a sparser attention mask as on the middle of Fig.2. For long video generation, it's necessary to attend history tokens to achieve temporal consistency. However, attending full history leads to an explosively long sequence. Considering each clip corresponds to 5s, which is sufficient to maintain temporal consistency, here we only attend to the last scale of the preceding clip. Finally, we obtain a highly sparse attention as show in Fig.2 (right). Our spacetime sparse attention drastically reduces attention computational overhead during both training and inference, all while delivering better performance.

## 4 Experiment

### 4.1 Implementation

**Datasets.** The training data of InfinityStar includes text-to-image data and text-to-video data. We curated 130M pretraining and 70M high-quality text-to-image data. To balance the data distribution and improve overall aesthetics, we also involve 5M high-quality synthetic data. In terms of text-to-video data, we curated around 16M video data. All videos are longer than 5 seconds. Among them 13M videos are under $336\times192$ resolution used for pre-training. They are mainly from Panda-70M[6],

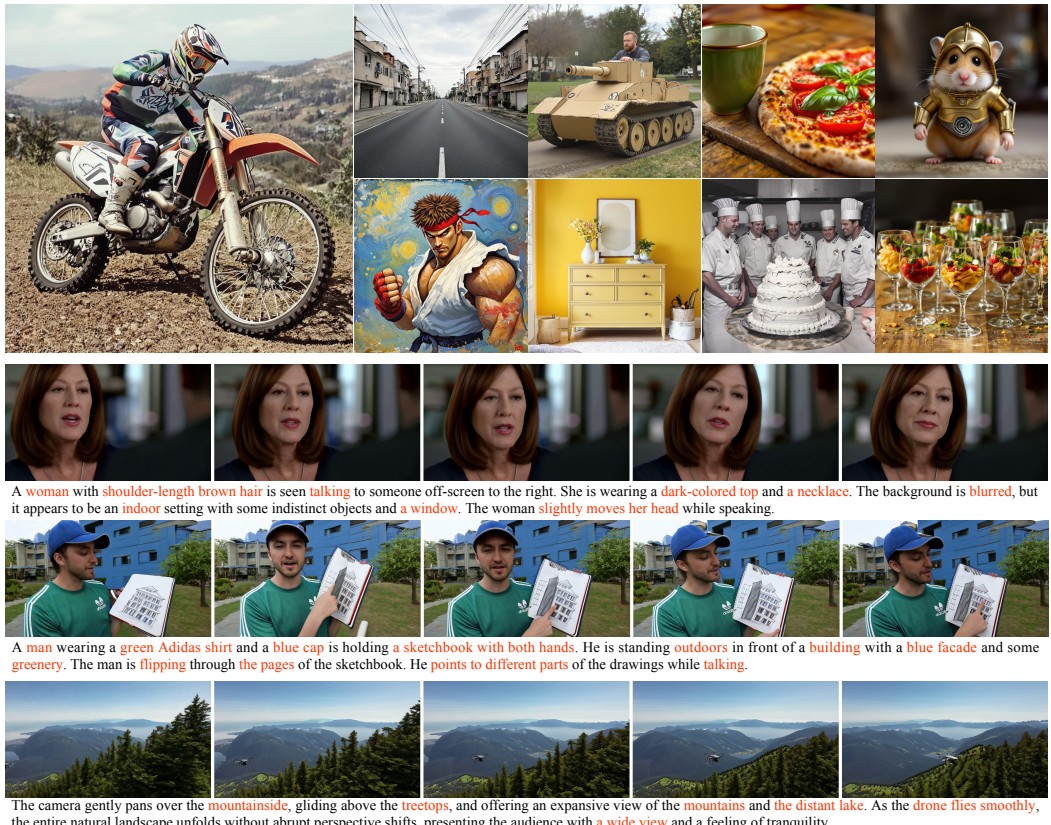

A woman with shoulder-length brown hair is seen talking to someone off-screen to the right. She is wearing a dark-colored top and a necklace. The background is blurred, but it appears to be an indoor setting with some indistinct objects and a window. The woman slightly moves her head while speaking.

A man wearing a green Adidas shirt and a blue cap is holding a sketchbook with both hands. He is standing outdoors in front of a building with a blue facade and some greenery. The man is flipping through the pages of the sketchbook. He points to different parts of the drawings while talking.

The camera gently pans over the mountainside, gliding above the treetops, and offering an expansive view of the mountains and the distant lake. As the drone flies smoothly, the entire natural landscape unfolds without abrupt perspective shifts, presenting the audience with a wide view and a feeling of tranquility.

Figure 3: Text to image and text to video examples.

Mira[14], and other internal video-text pairs. Apart from those 192p videos, we also curated 3M 480p and 50K 720p high-quality videos for fine-tuning.

**Model and Training.** After inserting the patchify and unpatchify layers between Wan 2.1 VAE's encoder and decoder, we obtain a video tokenizer with a compression rate of $4 \times 16 \times 16$ and a latent dimension of 64. Multi-scale BSQ quantization is adopted to obtain discrete tokens. In contrast to using a vocabulary size of $2^{64}$ for all scales, we use a vocabulary size of $2^{16}$ for the former small scales and $2^{64}$ for the latter large scales. We empirically find that it boosts convergence and has a negligible impact on the reconstruction quality. Starting with the pretrained weights of Wan 2.1 VAE, the discrete tokenizer is fine-tuned jointly on images of $256 \times 256$, $512 \times 512$, $768 \times 768$ and videos of $256 \times 256 \times 81$ for 30K iterations. The learning rate is $1e^{-4}$.

The autoregressive Transformer of InfinityStar is trained progressively in four stages, including a T2I pre-training and three T2V fine-tuning on 192p, 480p, 720p respectively. Each time we increase the training resolution, we preserve scale schedule of lower resolutions and append several larger scales, which enables better inheritance. The global batch size for 192p is 2048 and that of 480p and 720p is 1024. The learning rate for 192p is $2e^{-4}$. Then we decay it to $1e^{-4}$ for 480p and 720p. We train the model on videos of 192p, 480p, 720p for 50K, 8K, 3K iterations, respectively. Specifically, each clip pyramid is composed of 80 frames at 16 fps, and the first $K_s = 12$ semantic scales are repeated by $N = 3$ times.

## 4.2 Text-to-Image Generation

The upper part of Fig.3 shows generated images from our InfinityStar-T2I model, showcasing InfinityStar's strength in generating high-fidelity and photo-realistic images across various categories and image styles. We also carry out the quantitative evaluation on the GenEval[11] and DPG[13] benchmarks. As in Tab.1, InfinityStar achieves the best overall score of 0.79 on the GenEval bench with a prompt rewriter. It's worth noting that InfinityStar exceeds Infinity by 6% on overall score. We attribute the significant improvement to the larger model size and the architectural innovations. On the DPG bench, InfinityStar reaches an overall score of 86.55, surpassing Infinity by 3.09%. These

Table 1: Evaluation on the GenEval [11] and DPG [13] benchmark. † result is with prompt rewriting.

| Methods | # Params | GenEval↑ | | | | DPG↑ | | |
|---|---|---|---|---|---|---|---|---|
| | | Two Obj. | Position | Color Attri. | **Overall** | Global | Relation | **Overall** |
| Diffusion Models | | | | | | | | |
| SDXL [20] | 2.6B | 0.74 | 0.15 | 0.23 | 0.55 | 83.27 | 86.76 | 74.7 |
| PixArt-Sigma [5] | 0.6B | 0.62 | 0.14 | 0.27 | 0.55 | 86.89 | 86.59 | 80.5 |
| SD3 (d=38) [9] | 8B | 0.89 | 0.34 | 0.47 | 0.71 | - | - | - |
| SANA-1.0 [29] | 1.6B | - | - | - | 0.66 | - | - | 84.8 |
| FLUX-dev [18] | 12B | - | - | - | 0.67 | - | - | 84.0 |
| FLUX-schnell [18] | 12B | - | - | - | 0.71 | - | - | 84.8 |
| AutoRegressive Models | | | | | | | | |
| LlamaGen [21] | 0.8B | 0.34 | 0.07 | 0.04 | 0.32 | | | 65.2 |
| Chameleon [22] | 7B | - | - | - | 0.39 | - | - | - |
| Show-o [30] | 1.3B | 0.80 | 0.31 | 0.50 | 0.68 | - | - | 67.5 |
| Emu3 [28] | 8B | 0.81† | 0.49† | 0.45† | 0.66† | - | - | 81.6 |
| Infinity [12] | 2B | 0.85† | 0.49† | 0.57† | 0.73† | 93.11 | 90.76 | 83.46 |
| **InfinityStar-T2I** | 8B | **0.90†** | **0.62†** | **0.67†** | **0.79†** | 91.68 | 91.87 | 86.55 |

quantitative results demonstrate InfinityStar's strong capabilities of image generation following users' prompts.

## 4.3 Text-to-Video Generation

In the lower part of Fig.3, we present the generated videos of InfinityStar regarding user prompts. The generated videos successfully capture the semantic information in user prompts while maintaining high aesthetics and visual quality. Especially for the second example in Fig.3, the generated video accurately restores the delicate movements of the characters flipping through sketchbooks, talking while pointing to different parts of the drawings. In Tab.2, we compare InfinityStar with leading diffusion and autoregressive approaches on VBench—a comprehensive video benchmark spanning 16 evaluation dimensions. Our model achieves an overall score of 83.74, outperforming all open-source autoregressive baselines by a substantial margin. Moreover, InfinityStar surpasses diffusion-based competitors such as OpenSora[39], CogVideoX[32], and HunyuanVideo[16]. These results demonstrate that, through its novel spacetime autoregressive design, InfinityStar not only pushes the capabilities of discrete autoregressive video models but also attains performance on par with—and in some cases superior to—state-of-the-art diffusion methods.

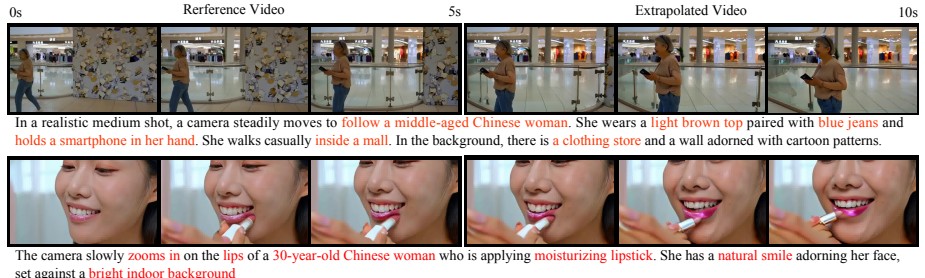

In a realistic medium shot, a camera steadily moves to follow a middle-aged Chinese woman. She wears a light brown top paired with blue jeans and holds a smartphone in her hand. She walks casually inside a mall. In the background, there is a clothing store and a wall adorned with cartoon patterns.

The camera slowly zooms in on the lips of a 30-year-old Chinese woman who is applying moisturizing lipstick. She has a natural smile adorning her face, set against a bright indoor background

Figure 4: Zero-shot video extrapolation examples.

**Zero-shot Generation.** Although trained exclusively on T2V data, InfinityStar can generate videos conditioned on an image or a video as historical without any fine-tuning. Fig.4 shows video extrapolation results. The synthesized videos exhibit strong temporal coherence with the reference while faithfully capturing the semantic nuances of texts. Zero-shot I2V samples are presented in the appendix.

## 4.4 Ablation Study

**Visual Tokenizer.** As shown in Fig.5 and Tab.3, loading weights of continuous video tokenizer significantly speeds up the convergence and achieves the best reconstruction results. As shown in Fig.6, stochastic quantizer depth largely improves the reconstruction quality of early scales. In terms of generation, using VAE with SQD leads to a notable improvement in VBench scores (81.28 *v.s.* 81.07 as shown in Tab.4). Moreover, we observe that SQD contributes to faster convergence during the video generation training.

Table 2: Evaluation on the VBench benchmark. † result is with prompt rewriting.

| Models | # Params | Human Action | Scene | Multiple Objects | Appear. Style | Quality Score | Semantic Score | Overall |
|---|---|---|---|---|---|---|---|---|
| Diffusion Models | | | | | | | | |
| AnimateDiff-V2 | 1.5B | 92.60 | 50.19 | 36.88 | 22.42 | 82.90 | 69.75 | 80.27 |
| VideoCrafter-2.0[4] | 1.5B | 95.00 | **55.29** | 40.66 | **25.13** | 82.20 | 73.42 | 80.44 |
| OpenSora V1.2[39] | 1.1B | 85.80 | 42.47 | 58.41 | 23.89 | 80.71 | 73.30 | 79.23 |
| Show-1[35] | 6B | 95.60 | 47.03 | 45.47 | 23.06 | 80.42 | 72.98 | 78.93 |
| Gen-3 [10] | - | 96.40 | 54.57 | 53.64 | 24.31 | 84.11 | 75.17 | 82.32 |
| CogVideoX-5B[32] | 5B | **99.40** | 53.20 | 62.11 | 24.91 | 82.75 | 77.04 | 81.61 |
| HunyuanVideo[16] | 13B | 94.40 | 53.88 | 68.55 | 19.80 | 85.09 | 75.82 | 83.24 |
| Wan 2.1[26] | 14B | 98.80 | 53.67 | **81.44** | 21.13 | **85.64** | **80.95** | **84.70** |
| AutoRegressive Models | | | | | | | | |
| Nova[8]† | 0.6B | 95.20 | 54.06 | 77.52 | 20.92 | 80.39 | 79.05 | 80.12 |
| Emu3[28] | 8B | 77.71 | 37.11 | 44.64 | 20.92 | 84.09 | 68.43 | 80.96 |
| **InfinityStar**† | 8B | 96.43 | 52.08 | 78.66 | 21.81 | 84.73 | 79.78 | 83.74 |

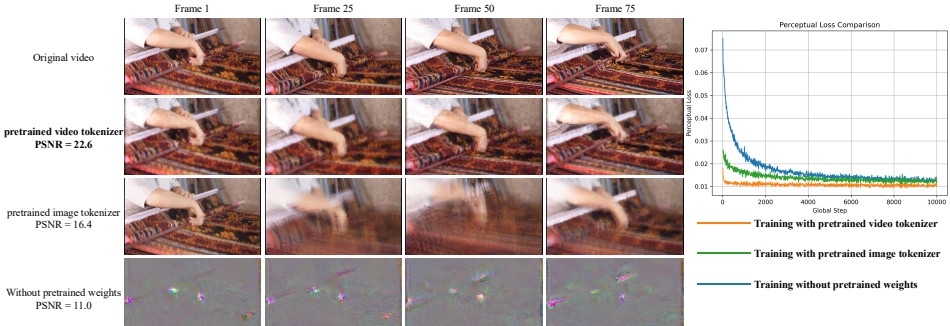

Figure 5: Influence of pretrained weights on reconstruction and convergence. The left sub-figure shows the reconstructed frames using different pretrained weights `without finetuning`. Loading weights of continuous video tokenizer achieves the best results. The right sub-figure shows that training with pretrained video tokenizer converges significantly faster than the other two strategies.

Table 3: Reconstruction metrics on an internal high-motion video benchmark (480p 81 frames).

| Pretrained Weights | PSNR($\uparrow$) | SSIM($\uparrow$) | LPIPS($\downarrow$) |
|---|---|---|---|
| Continuous Video VAE | **33.37** | **0.94** | **0.065** |
| Image VAE | 29.10 | 0.90 | 0.123 |
| None | 30.04 | 0.90 | 0.124 |

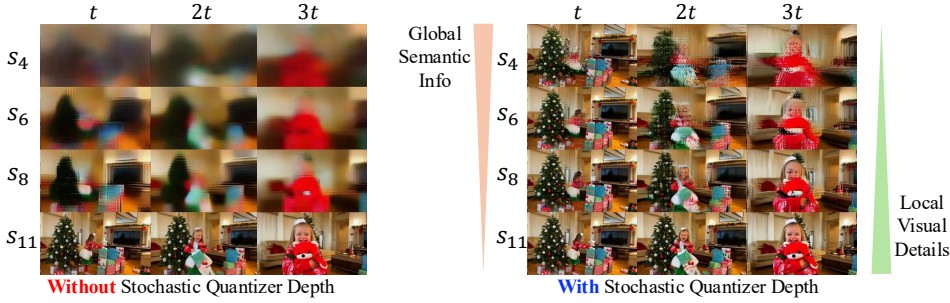

Figure 6: The influence of stochastic quantizer depth. Sub-figure $(s_i, nt)$ represents the reconstructed frame $nt$ using all tokens from the image pyramid plus tokens of first $i$ scales in the clip pyramid. SQD significantly improves the reconstruction quality of early scales. Besides, the earlier scales correspond to global semantics, while the later ones are responsible for local visual details.

**Pseudo-Spacetime Pyramid *v.s.* Spacetime Pyramid.** As illustrated in Fig.7, videos generated by the pseudo-spacetime pyramid lack visual details and deliver simpler motion. In contrast, spacetime pyramid generates videos with richer details and higher motion. Besides, spacetime pyramid improves VBench's overall score from 80.30 to 81.28 as illustrated in Tab.4. These experiments support the hypothesis that spacetime pyramid could decouple appearance and temporal information. The image

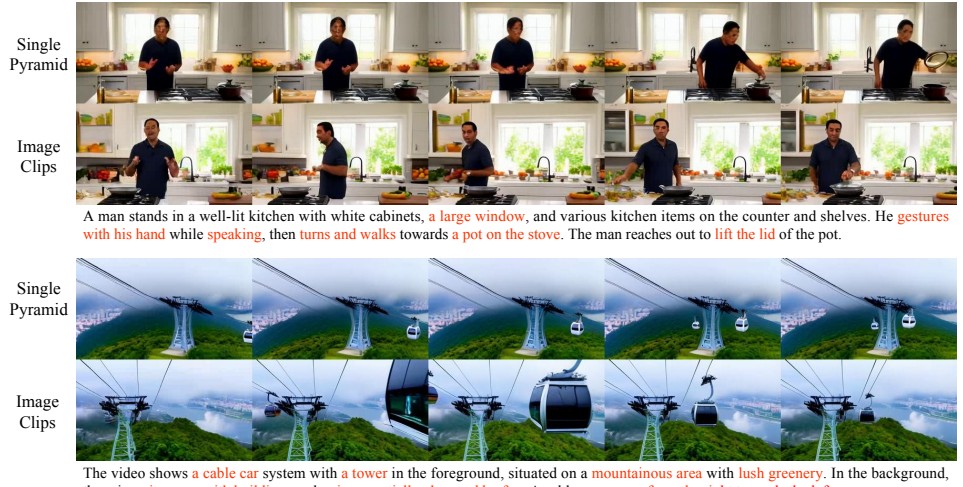

Single Pyramid

Image Clips

A man stands in a well-lit kitchen with white cabinets, a large window, and various kitchen items on the counter and shelves. He gestures with his hand while speaking, then turns and walks towards a pot on the stove. The man reaches out to lift the lid of the pot.

Single Pyramid

Image Clips

The video shows a cable car system with a tower in the foreground, situated on a mountainous area with lush greenery. In the background, there is a cityscape with buildings and a river, partially obscured by fog. A cable car moves from the right towards the left.

Figure 7: Comparison between Pseudo-Spacetime Pyramid and Spacetime Pyramid. Spacetime Pyramid could generate videos with richer details and higher motion.

pyramid corresponds to the appearance information and clip pyramids focus on subsequent motions. This decoupling makes it easier to learn video motions. In addition to advances in performance, spacetime pyramid could unify T2I, T2V, I2V tasks into one framework.

**Semantic Scale Repetition.** In Fig.6, we can observe that the earlier scales correspond to semantic information, while the later ones are responsible for high-frequency details. Here we compare the generation results with and without semantic scale repetition. As shown in Fig.8, semantic scale repetition is highly effective in improving the structure stability and motion quality. The quantitative results further confirm the significant gains. As shown in Tab.4, semantic scale repetition improves VBench's overall score from 75.72 to 81.28.

**Spacetime Sparse Attention.** In Tab.4 and Tab.5, we compare different attention mechanisms. Spacetime sparse attention shows superior performance to full attention in the Vbench total score (81.28 *v.s.* 80.77), while showing a significant advantage in saving computation and GPU VRAM. SSA reaches $1.5\times$ speedup when generating 192p 161 frames. The efficiency advantage becomes larger as the resolution and duration grow. For 480p 161 frames, full attention fails due to OOM while SSA completes it within 44.7s using 63GB VRAM. We hypothesize that SSA produces better results than full attention because it reduces exposure bias. Full attention is more susceptible to accumulated errors. The reason we do not condition on smaller scales of the preceding clip is that it misses the former clips' visual details and brings visual inconsistency between clips. Although it reaches $1.1\times$, $1.5\times$ speedup for 192p and 480p 161 frames, we observe a significant performance drop in Vbench from 81.28 to 80.75 as shown in Tab.4. Therefore, the proposed spacetime sparse attention strikes a better balance between computational efficiency and visual quality.

### 4.5 Inferency Latency

As shown in Tab.6, we report the end-to-end inference latency measured on a single GPU, including both the text encoder and VAE decoder. Wan-2.1[26] and Nova[8] were evaluated using their default GitHub configurations. Even without employing stronger compression, InfinityStar achieves a $32\times$ speedup over Wan-2.1. Furthermore, despite its larger model size, InfinityStar delivers a $6\times$ speedup compared to Nova. These results highlight our model's significant efficiency advantage over both diffusion and autoregressive approaches.

## 5 Conclusion and Limitation

We introduce InfinityStar, a unified spacetime autoregressive framework capable of synthesizing high-resolution images and dynamic, high-motion videos. By seamlessly integrating spatial and temporal prediction within a purely discrete architecture, InfinityStar supports diverse generation tasks while maintaining both state-of-the-art quality and exceptional efficiency. Our extensive evaluation

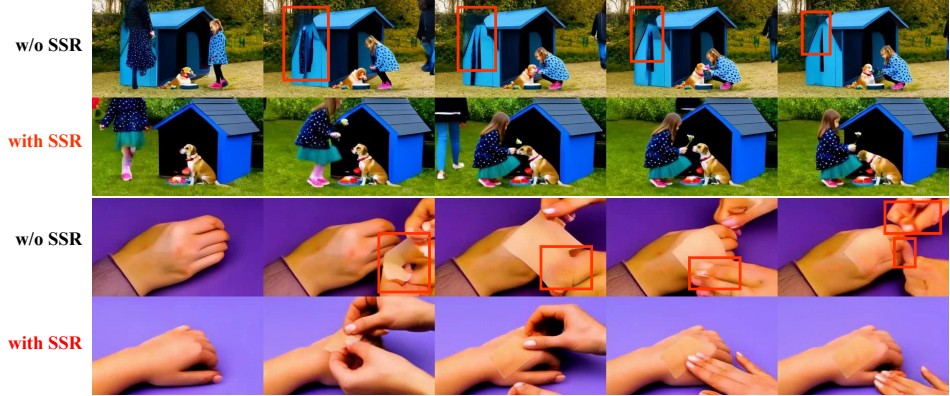

Figure 8: Semantic Scale Repetition (SSR) greatly improves structure stability and motion quality.

Table 4: Comprehensive ablation studies. Experiment with 1M 192p training data, $batch\_size = 40$, and 30K iterations. We evaluate the results on the Vbench benchmark.

| Vbench | total score | quality score | semantic score |
|---|---|---|---|
| **InfinityStar (Our Model)** | **81.28** | **81.56** | 80.16 |
| *Attend to former clip's largest scale* | | | |
| *Ablation by removing/replacing core components* | | | |
| w/o Semantic Scale Repetition(SSR) | 75.72 | 76.73 | 71.68 |
| w/o Spacetime Pyramid (using Pseudo-Spacetime) | 80.30 | 80.81 | 78.28 |
| w/o Stochastic Quantizer Depth(SQD) | 81.07 | 81.21 | **80.54** |
| *Comparison of different Attention Mechanism variants* | | | |
| Full Attention | 80.77 | 81.15 | 79.23 |
| Attend to former clip's 3rd largest scale | 80.86 | 81.26 | 79.26 |
| Attend to former clip's 6th largest scale | 80.75 | 80.98 | 79.80 |

Table 5: Computational efficiency comparison of attention mechanisms on a single GPU.

| | (192p 65 frames) | (192p 161 frames) | (480p 161 frames) |
|---|---|---|---|
| Full Attention | 8.6s / 40.8GB | 24.3s / 57GB | OOM |
| Attend to former clip's largest scale | 7.7s / 38.5GB | 16.7s / 40GB | 44.7s / 63 GB |
| Attend to former clip's 3rd largest scale | 7.4s / 38.2GB | 15.8s / 39GB | 34.5s / 58 GB |
| Attend to former clip's 6th largest scale | 7.3s / 37.9GB | 15.2s / 38GB | 30.5s / 55GB |

Table 6: Computational efficiency comparison.

| Method | Model | # Parameters | Durations(s) | Frames | Resolution | Time(s) | Speedup |
|---|---|---|---|---|---|---|---|
| Diffusion | Wan 2.1[26] | 14B | 5 | 81 | 720p | 1864 | 1 |
| AR | Nova[8] | 0.6B | 5 | 81 | 480p | 354 | 5 |
| AR | InfinityStar | 8B | 5 | 81 | 720p | 58 | 32 |

demonstrates that InfinityStar outperforms prior autoregressive video models and rivals leading diffusion-based approaches, producing a 720p video of 5s in one-tenth the inference time. As the first discrete autoregressive model to deliver industrial-grade 720p video synthesis, we anticipate that InfinityStar will catalyze future research on rapid, long video generation.

While InfinityStar sets a new record in discrete video generation, several limitations remain. Specifically, there is a trade-off between image quality and motion fidelity in high-motion scenes, where sometimes fine-grained visual details can be compromised. Additionally, due to limited computational resources, we have not scaled our model training or parameter size to match those of leading diffusion models, which constrains the upper bound of the performance. Furthermore, our inference pipeline has not yet been fully optimized, indicating room for future improvement.

# 6 Acknowledgments

The authors appreciate the valuable support provided by colleagues from ByteDance, including Yuqi Zhang, Yifu Zhang, Hao Yang, Yifei Hu, Chuang Lin, Xiaofeng Mei, Ruibiao Lu, and Jiawei Duan. Their contributions to data processing and related technical aspects are essential for the advancement of this research.

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
