# OpenReview forum: "InfinityStar: Uniﬁed Spacetime AutoRegressive Modeling for Visual Generation"
_NeurIPS.cc/2025/Conference — NeurIPS 2025 oral_

### Official Review · Reviewer_KwMu · 2025-06-30

**Clarity:** 3
**Significance:** 4
**Originality:** 3
**Rating:** 5
**Confidence:** 5

**Summary:**

This paper proposes Infinity∗, a spacetime pyramid modeling framework that unifies diverse visual generation tasks, including t2i, t2v, and i2v. It systematically investigates the various challenges encountered when applying the Infinity architecture to video generation and proposes solutions for each issue. The method elevates the performance of autoregressive video generation approaches to a level comparable to cutting-edge diffusion models.

**Questions:**

1. How to determine the video generation length, frame count, or when to stop generation.
2. Does Spacetime Sparse Attention affect performance?
3. In Table 4, it is better to list the frame numbers instead of durations to ensire comparative fairness.

**Ethical Concerns:**

["NO or VERY MINOR ethics concerns only"]

**Final Justification:**

Thanks for the author's response. Most of my concerns are resolved. I keep the original rating unchanged.

**Limitations:**

Yes

**Paper Formatting Concerns:**

No paper formatting concerns.

**Quality:**

4

**Strengths And Weaknesses:**

Strengths:
1. Infinity∗ is a vision generation model based on the Infinity architecture, covering a variety of visual generation tasks. Notably, in video generation, it significantly outperforms other autoregressive text-to-video models of the same category, and the experimental results are solid.
2. The paper is well-written and clear. Starting from Infinity, it explains the challenges and solutions of the VAR architecture in visual generation.
3. Overall, although it is an extension of Infinity, it also holds considerable value in unifying diverse visual generation tasks, providing inspiration for future methods in autoregressive visual generation.

Weaknesses:
1. Some content in Section 3.2 is confusing. For example, the "Pyramid" in Figure 1 still appears to be a spatial pyramid at each timestep, making the term "spacetime pyramid" misleading.
2. Some experimental details are insufficient. Are there any experiments demonstrating that Spacetime Sparse Attention has minimal impact on performance?

---

> ### Author Rebuttal · Authors · 2025-07-31
>
> ## Response to Reviewer ``KwMu`` (Rating: Accept)
>
> We thank reviewers for your valuable comments on Infinity$^*$, acknowledging strong performance with fast speed (``7YkR``,
> ``qR4k``, ``vUvY``, ``tNTA``, ``KwMu``), well-motivated design (``7YkR``, ``vUvY``, ``KwMu``), valuable contribution to the generative model community (``qR4k``, ``tNTA``, ``KwMu``), with solid experimental results (``qR4k``, ``KwMu``). Your advice will help us enhance the paper.
>
> #### **Q1:  Some content in Section 3.2 is confusing. For example, the ''Pyramid" in Figure 1 still appears to be a spatial pyramid at each timestep, making the term ''spacetime pyramid" misleading.**
>
> Thank you for your insightful comment. We named our scale schedule the "spacetime pyramid" for two primary reasons: (1) To distinguish it from the conventional space pyramid schedule used in image generation. Specifically, images are two-dimensional, involving only height and width. Consequently, the scale schedule in images consists of a series of coordinate pairs $(h_i, w_i)$. Videos, however, are a spatiotemporal medium with three dimensions (time, height, and width). Accordingly, the new schedule we propose is composed of triplets $(t_i, h_i, w_i)$. The term "spacetime" is intentionally used to emphasize that our method is designed to process videos with their inherent spatiotemporal structure.
>     (2) To reflect the hierarchical nature of our schedule, which is the core characteristic of a pyramid. Although the temporal dimension t remains constant across the different scales in our schedule, we observe a clear hierarchical pattern, as illustrated in Figure 6 of the submitted manuscript. The earlier scales encode global semantic information, while the later scales focus on local, fine-grained details. As one progresses from the initial to the final scales, semantics gradually decreases while details become progressively stronger. Since this hierarchical structure is the most fundamental attribute of a pyramid, we believe the term "pyramid" accurately describes our approach.
>
> #### **Q2:  any experiments demonstrating that Spacetime Sparse Attention has minimal impact on performance?**
> In Tables 1~2, we compare our spacetime sparse attention with full attention. Spacetime sparse attention shows better performance than full attention in the Vbench's score (81.28 vs 80.77), while showing a significant advantage in saving computation and GPU VRAM.  We conclude that spacetime sparse attention significantly improves efficiency without performance loss. We hypothesize that spacetime sparse attention yields better results than full attention because it reduces exposure bias. In full attention, since the context is all ground truth during training, it is more susceptible to accumulated errors during inference.
>
> | Method                                    | total score | quality score | semantic score |
> |---|:---:|:---:|:---:|
> | Full Attention                                 | 80.77       | 81.15         | 79.23          |
> | Attend to former clip's largest scale     | **81.28**       |  **81.56**         | **80.16**          |
> | Attend to former clip's 3rd largest scale | 80.86       | 81.26         | 79.26          |
> | Attend to former clip's 6th largest scale | 80.75       | 80.98         | 79.80          |
>
> Table 1. Performance comparison of various attention mechanisms
>
> ---
>
> |           Method                   | (192p 65 frames, 4s 16fps) | (192p 161 frames, 10s 16fps) | (480p 161 frames, 10s 16fps) |
> |---|:---:|:---:|:---:|
> | Full Attn                               | 8.6s / 40.8GB              | 24.3s / 57GB                 | OOM                          |
> | Attend to former clip's largest scale   | 7.7s / 38.5GB              | 16.7s / 40GB                 | 44.7s / 63GB                 |
> | Attend to former clip's 3rd largest scale | 7.4s / 38.2GB              | 15.8s / 39GB                 | 34.5s / 58GB                 |
> | Attend to former clip's 6th largest scale | 7.3s / 37.9GB              | 15.2s / 38GB                 | 30.5s / 55GB                 |
>
> Table 2. Computational efficiency comparison of attention mechanisms measured on a single NVIDIA H100 GPU with 8B parameters.
>
> #### **Q3:  How to determine the video generation length, frame count, or when to stop generation?**
>
> During the training phase, variable-length videos are decomposed into clips, where each clip contains 80 frames (5s 16FPS). The caption regarding each video is appended with ''duration=xs". During the inference phase, Infinity* stops generating once the user-specified duration is reached.
>
> #### **Q4:  In Table 4, it is better to list the frame numbers instead of durations to ensire comparative fairness.**
>
> Thanks for your valuable suggestion. In Table 4 in the paper, all compared models generate 16 FPS videos with 81 frames. We will replace ''duration" with ''frames numbers" in the revised paper.
>
> #### **Experiments Setup**
> All our ablation experiments were conducted using a 1M training set with a batch size of 40, trained at 192p resolution for 30,000 iterations. We evaluated the results on the Vbench benchmark.

---

### Official Review · Reviewer_tNTA · 2025-07-01

**Clarity:** 3
**Significance:** 3
**Originality:** 2
**Rating:** 5
**Confidence:** 4

**Summary:**

In this work, the authors propose Infinity*, a pyramid modeling autoregressive (AR) model for image and video generation. Infinity* builds a discrete vision tokenizer that tokenizes videos into discrete tokens in a pseudo-spacetime pyramid. In particular, each video is encoded into a a sequential tokens that models video clips of different scales. An AR model is trained on the pseudo-spacetime pyramid tokenizer with spacetime attention mask for efficient generation and time constant generation. Infinity* demonstrates competitive performance on text-to-image and text-to-video generation benchmarks. Also, it experimentally shows effectiveness on design choices like stochastic quantizer depth and semantic scales repetition.

**Questions:**

Major:
1. The work mentioned pre-trained video VAE with continuous latents is used. How is the VAE pretrained? Is it from previous open-source work or trained from scratch? More details about the VAE are appreciated.
2. For comparison of different models in Table 1/2, is it possible to include the scale of training data besides # params so that people can better compare Infinity* with baselines.
3. Also in Table 2, it would be beneficial if # params of different models are listed as in Table 1.
4. Are there quantitative ablation studies about the proposed stochastic quantizer depth and semantic scales repetition as well as how the hyperparameters are determined? For example, number of repetitions for semantic scales.

Minor:
1. In page 5 line 197, it should be "Patchify" not "Patchfy".

**Ethical Concerns:**

["NO or VERY MINOR ethics concerns only"]

**Final Justification:**

The work has shown strong performance empirically with comprehensive experiments supporting the design choices after rebuttal. I believe this work can benefit building efficient AR-based video generation.

**Limitations:**

yes

**Quality:**

3

**Strengths And Weaknesses:**

Strengths:
1. The paper is well written and easy to follow.
2. The work shows how AR model can achieve competitive performance in text-to-video generation, which is a valuable contribution to generative model community.
3. The work shows competitive performance on vision generation benchmarks.

Weaknesses:
1. Overall, the paper is largely built on previous VAR and Infinity and shows design choices that makes scale-wise AR model work for video generation. Though design choices like stochastic quantizer depth and semantic scales repetition help improve the generation quality, they may be adhered to certain AR model instead of benefiting vision generative models in general.

---

> ### Author Rebuttal · Authors · 2025-07-31
>
> ## Response to Reviewer ``tNTA`` (Rating: Borderline Accept)
>
> We thank reviewers for your valuable comments on Infinity$^*$, acknowledging strong performance with fast speed (``7YkR``,
> ``qR4k``, ``vUvY``, ``tNTA``, ``KwMu``), well-motivated design (``7YkR``, ``vUvY``, ``KwMu``), valuable contribution to the generative model community (``qR4k``, ``tNTA``, ``KwMu``), with solid experimental results (``qR4k``, ``KwMu``). Your advice will help us enhance the paper.
>
> #### **Q1:   The paper is largely built on previous VAR and Infinity and shows design choices that makes scale-wise AR model work for video generation. Though design choices like stochastic quantizer depth (SQD) and semantic scales repetition (SSR) help improve the generation quality, they may be adhered to certain AR model instead of benefiting vision generative models in general.**
>
> Thanks for your suggestions. We agree that Infinity* is built on VAR and Infinity. However, both VAR and Infinity focus on image generation. Simply extending them to video generation yields inferior results, and there is currently no relevant work in academia. In this paper, we successfully solved the difficulty of handling the additional temporal dimension while preserving high efficiency. Compared to vanilla AR and diffusion models, Infinity$^*$ reaches 10$\sim$30$\times$ speedup and better generation quality. Therefore, we believe our work is non-trivial in fostering better and faster video generation.
>
> As for the universality of SQD and SSR, we agree that they are designed for VAR video generation and may not be suitable for vanilla AR models. This lies in the hierarchical structure of VAR, which is different to vanilla AR. But these innovations are applicable to other hierarchical generation models, like FlowAR and FractalAR. We will explore applying SQD and SSR to other models.
>
> #### **Q2:  The work mentioned pre-trained video VAE with continuous latents is used. How is the VAE pretrained? Is it from previous open-source work or trained from scratch? More details about the VAE are appreciated.**
> Thank you for pointing this out. We chose the powerful open-sourced WAN 2.1 VAE as the continuous VAE. It is a 3D causal VAE with a latent dimension of 16, compressing videos by 4x8x8 times. As mentioned in the Wan2.1 paper, Wan-VAE is progressively trained in three stages: (1) image pretraining (2) low-resolution, short video pretraining (3) high-quality video fine-tuning. In Infinity$^*$, we construct our discrete tokenizer by inserting our quantizer between the encoder and the decoder of Wan-VAE and initialize it using the official weights of Wan-VAE. Then we fine-tune it on images and videos jointly. With the help of pretrained weights of continuous VAE, the convergence speed of the discrete tokenizer has been significantly improved as shown in Figure 5 of the submitted manuscript. We will clarify this in the revised paper.
>
> #### **Q3:  For comparison of different models in Table 1/2, is it possible to include the scale of training data besides params so that people can better compare Infinity\* with baselines.**
> Thanks for your advice. As shown in Table 1, we list the training data size for different T2I and T2V models if their original papers have declared. Wanx 2.1 and HunyuanVideo have not provided the information. We will add the training data size to Table 1/2 in the revised paper.
>
> ---
> |   T2I Model |   #Param  |   Data Size   |   T2V Model   |    #Param   |   Data Size   |
> |---|:---:|:---:|---|:---:|:---:|
> | PixArt-Sigma | 0.6B | 33M | VideoCrafter2 | 2B | 10M |
> | LlamaGen | 2.6B | 60M | OpenSora v1.2 | 1B | 30M |
> | SD3 | 8B | 1000M | NOVA | 0.6B | 20M |
> | Chameleon | 7B | 1400M | HunyuanVideo | 2B | undeclared |
> | Infinity | 2B | 130M | Wan 2.1 | 2B | undeclared |
> | Infinity*| 8B | 70M | Infinity* | 2B | 13M |
>
> Table 1. The number of parameters and training data size for different T2I/T2V models
>
> #### **Q4:  Also in Table 2, it would be beneficial if params of different models are listed as in Table 1.**
> We agree with your suggestion. As illustrated in Table 1, we present the number of parameters for open-source T2I/T2V models. Specifically, our model has 8B parameters, WAN 2.1 has 14B, HunyuanVideo has 13B, NOVA has 0.6B, and Emu3 has 8B parameters. We will add this information to the revised paper.
>
>  #### **Q5: Are there quantitative ablation studies about the proposed stochastic quantizer depth and semantic scales repetition as well as how the hyperparameters are determined?**
>
> Thank you for your valuable feedback. We provide quantitative ablation studies on both the stochastic quantizer depth (SQD) and semantic scales repetition (SSR) in Table 2 and Table 3, respectively. As shown in Table 2, incorporating SQD leads to a notable improvement in VBench scores across several metrics. Moreover, we observe that SQD contributes to faster convergence during video generation training: at 5k iterations, the variant with SQD achieves a gFVD of 257.03, compared to 349.46 without SQD. Regarding SSR, Table 3 demonstrates that introducing this mechanism improves the VBench score from 75.72 to 80.97. Increasing the number of repetition steps further boosts the score to 81.28, although with diminishing gains. Based on this trade-off between performance and computational efficiency, we set the repetition count to $n=6$.
>
> ---
> | Method   | total score | quality score | semantic score |
> |---|:---:|:---:|:---:|
> | w/o SQD  | 81.07       | 81.21         | **80.54**          |
> | With SQD | **81.28**       | **81.56**         | 80.16          |
>
> Table 2. Ablation study of the proposed stochastic quantizer depth
>
>
>
>
> ---
> | Method |  total score  |  quality score  |   semantic score   |
> |---|:---:|:---:|:---:|
> | w/o SSR | 75.72 | 76.73 | 71.68 |
> | With SSR (repetition=4) | 80.97 | 81.38 | 79.32 |
> | With SSR (repetition=6) | **81.28** | **81.56** | **80.16** |
>
> Table 3. Comparison of different repetition times in Semantic Scale Repetition (SSR)
>
>
>  #### **Q6: On page 5 line 197, it should be ''Patchify'' not ''Patchfy''.**
>
> Thank you for pointing out the typo on page 5, line 197. We appreciate your careful review. We will correct ''Patchfy'' to ''Patchify" in the revised paper.
>
>
> #### **Experiments Setup**
> All our ablation experiments were conducted using a 1M training set with a batch size of 40, trained at 192p resolution for 30,000 iterations. We evaluated the results on the Vbench benchmark.

---

> ### Comment · Reviewer_tNTA · 2025-08-03
> **Official Comment by Reviewer tNTA**
>
> I thank the authors for answering my questions. I believe the work has shown competitive performance in video generation empirically, and the additional experimental results in rebuttal further validated the design choices. I stay positive about the work and have increased my rating to 5.

---

> > ### Author Response · Authors · 2025-08-05
> >
> > Thank you very much for your positive feedback and for increasing your rating to 5. We truly appreciate your thoughtful comments and suggestions.

---

### Official Review · Reviewer_vUvY · 2025-07-02

**Clarity:** 4
**Significance:** 4
**Originality:** 4
**Rating:** 5
**Confidence:** 4

**Summary:**

This paper proposes a unified spacetime autoregressive framework for high-resolution image and dynamic video synthesis. Building on the success of VAR models, it models videos as an image pyramid and multiple clip pyramids, naturally decoupling static appearance from dynamic motion. Through key improvements including knowledge inheritance from continuous video tokenizers, stochastic quantizer depth, and semantic scale repetition, the model achieves  a good result on VBench, outperforming all VAR model and be comparible with Diffusion models.

**Questions:**

see weakness.

**Ethical Concerns:**

["NO or VERY MINOR ethics concerns only"]

**Final Justification:**

Authors solved my concerns, I think this paper is well contributed and have a great novelty and impact, I suggest to accept.

**Limitations:**

see weakness.

**Paper Formatting Concerns:**

no concern

**Quality:**

4

**Strengths And Weaknesses:**

## Strengths
1. It is so impressive to see such a great dynamic degree on VBench. And achieve compariable performance as SOTA on VBench but get a much faster speed.
2. Decomposes videos into image pyramids and clip pyramids, cleverly decoupling appearance and motion, enabling unified handling of T2I, T2V, I2V tasks.
3. By inheriting weights from pretrained continuous video VAE, significantly accelerates discrete tokenizer convergence and improves reconstruction quality.
4. Stochastic Quantizer Depth to address imbalanced information distribution, Semantic Scale Repetition to improve structural stability and motion quality, and Spacetime Sparse Attention to reduce computational overhead. The design of these module is novel and useful.

## Weaknesses
1. Authors mentioned that due to GPU and training data constraints, they didn't match the training scale of leading diffusion models like Wan 2.1, potentially limiting current performance.
2. Experiments mainly focus on short videos, lacking validation of longer temporal generation capabilities.
3. Performance still lags behind some diffusion models in certain evaluation dimensions (it is a weakness, but it is already very good for the VAR model, with considering a much faster generation speed.)

---

> ### Author Rebuttal · Authors · 2025-07-31
>
> ## Response to Reviewer ``vUvY`` (Rating: Accept)
>
> We thank reviewers for your valuable comments on Infinity$^*$, acknowledging strong performance with fast speed (``7YkR``,
> ``qR4k``, ``vUvY``, ``tNTA``, ``KwMu``), well-motivated design (``7YkR``, ``vUvY``, ``KwMu``), valuable contribution to the generative model community (``qR4k``, ``tNTA``, ``KwMu``), with solid experimental results (``qR4k``, ``KwMu``). Your advice will help us enhance the paper.
>
>
> #### **Q1:   Authors mentioned that due to GPU and training data constraints, they didn't match the training scale of leading diffusion models like Wan 2.1, potentially limiting current performance.**
> Yes, it is one of the limitations of our work. We have not yet matched the training scale of leading diffusion models like Wan 2.1 in terms of GPU resources and data. By scaling up both, our results shall keep improving.
>
> #### **Q2:  Experiments mainly focus on short videos, lacking validation of longer temporal generation capabilities.**
> Thank you for the insightful comment. We acknowledge that the experiments in the submitted manuscript primarily focus on short video clips (less than 10 seconds). This is largely due to the limited availability of datasets containing long videos paired with rich, multi-segment captions. Nevertheless, our autoregressive framework is inherently more suitable for long video generation than diffusion-based models, which typically operate with fixed-length sequences. In future work, we plan to curate more appropriate data and extend our approach to thoroughly evaluate its capabilities in long video generation.
>
> #### **Q3: Performance still lags behind some diffusion models in certain evaluation dimensions(it is a weakness, but it is already very good for the VAR model, with considering a much faster generation speed.)**
> Thank you for acknowledging our results. We agree that our method currently lags behind some leading diffusion models in certain aspects. Our work explores discrete video generation based on VAR and Infinity, opening up new possibilities in video generation field. While diffusion models have been extensively studied, we will continue to improve our approach to bridge the gap while maintaining our speed advantage.

---

> > ### Comment · Reviewer_vUvY · 2025-08-05
> >
> > Thanks for the author's serious and responsible response, which solved my concerns. I am willing to keep the rating.

---

### Official Review · Reviewer_qR4k · 2025-07-02

**Clarity:** 3
**Significance:** 3
**Originality:** 2
**Rating:** 5
**Confidence:** 3

**Summary:**

The authors present Infinity-star, a discrete autoregressive framework that unifies text-to-image, text-to-video, image-to-video and zero-shot video extrapolation. It extends the “next-scale” paradigm of Infinity to the temporal domain by decomposing a video into an image pyramid plus successive clip pyramids that grow only in space, allowing static appearance and motion to be modeled separately. The introduced model achieve competitive performance on benchmarks of several tasks. More importantly, the proposed approach significantly reduces inference speed compared to existing diffusion models.

**Questions:**

- It would be helpful if the author could list the sequence length of each sample (summing all scales) during training.

**Ethical Concerns:**

["NO or VERY MINOR ethics concerns only"]

**Final Justification:**

After reading other review and the author rebuttal, i believe this is a good paper and I don't have any major concerns. Therefore, I keep my initial score.

**Limitations:**

Yes

**Quality:**

3

**Strengths And Weaknesses:**

## Strengths:
- The proposed model achieve fairly competitive results with much faster inference speed compared to diffusion/flow matching.
- The paper is well-written, organized and technically sound. Several design details are included for reproducibility.
- The author validate their motivation for extending spatial infinity to temporal dimension with simple baselines. They also report ablation study to demonstrate the effectiveness of proposed approach.
- The open-source weight would be beneficial and have large impact for community

## Weakness:
- The author experiment with several design choices and chose the best one but only report qualitative results. It would be helpful if they can also report quantitative results for a clearer picture for reader i.e., to understand how significant of each component that the paper introduce.

---

> ### Author Rebuttal · Authors · 2025-07-31
>
> ## Response to Reviewer ``qR4k`` (Rating: Accept)
>
> We thank reviewers for your valuable comments on Infinity$^*$, acknowledging strong performance with fast speed (``7YkR``,
> ``qR4k``, ``vUvY``, ``tNTA``, ``KwMu``), well-motivated design (``7YkR``, ``vUvY``, ``KwMu``), valuable contribution to the generative model community (``qR4k``, ``tNTA``, ``KwMu``), with solid experimental results (``qR4k``, ``KwMu``). Your advice will help us enhance the paper.
>
>
> #### **Q1:  The author experiment with several design choices and chose the best one but only report qualitative results. It would be helpful if they can also report quantitative results for a clearer picture for reader i.e., to understand how significant of each component that the paper introduce.**
> We appreciate your suggestion to provide more quantitative results. In response, we have included additional ablation studies focusing on the proposed spacetime sparse attention, stochastic quantizer depth, semantic scales repetition, and Spacetime Pyramid  in Tables 1~4, respectively. The results further validate the contribution of each component to the overall performance of our method.
>
> ---
>
> | Method                                    | total score | quality score | semantic score |
> |---|:---:|:---:|:---:|
> | Full Attention                                 | 80.77       | 81.15         | 79.23          |
> | Attend to former clip's largest scale     | **81.28**       | **81.56**         | **80.16**          |
> | Attend to former clip's 3rd largest scale | 80.86       | 81.26         | 79.26          |
> | Attend to former clip's 6th largest scale | 80.75       | 80.98         | 79.80          |
>
> Table 1. Performance comparison of various attention mechanisms
>
> ---
>
> | Method |  total score  |  quality score  |   semantic score   |
> |---|:---:|:---:|:---:|
> | w/o SSR | 75.72 | 76.73 | 71.68 |
> | With SSR (repetition=4) | 80.97 | 81.38 | 79.32 |
> | With SSR (repetition=6) | **81.28** | **81.56** | **80.16** |
>
> Table 2. Comparison of different repetition times in Semantic Scale Repetition (SSR)
>
> ---
>
> | Method   | total score | quality score | semantic score |
> |---|:---:|:---:|:---:|
> | w/o SQD  | 81.07       | 81.21         | **80.54**          |
> | With SQD | **81.28**       | **81.56**         | 80.16          |
>
> Table 3. Ablation study of the proposed stochastic quantizer depth
>
> ---
>
> | Method                 | total score | quality score | semantic score |
> |---|:---:|:---:|:---:|
> | Pseudo-Spacetime Pyramid | 80.30       | 80.81         | 78.28          |
> | Spacetime Pyramid      | **81.28**       | **81.56**         | **80.16**          |
>
> Table 4. Ablation study of the proposed Spacetime Pyramid
>
> #### **Q2:  It would be helpful if the author could list the sequence length of each sample (summing all scales) during training.**
>
> As shown in Table 5, we list the total sequence length for each sample by summing across all scales at different resolutions during training.
>
> |  Resolution | seq_len (81 frames) | seq_len (161 frames) |
> |:---:|:---:|:---:|
> | 192p      | 14,721              | 28,741               |
> | 384p      | 48,489              | 94,669               |
> | 480p      | 81,879              | 159,859              |
> | 720p      | 205,863             | 401,923              |
>
> Table 5. sequence length of training sample
>
> #### **Experiments Setup**
> All our ablation experiments were conducted using a 1M training set with a batch size of 40, trained at 192p resolution for 30,000 iterations. We evaluated the results on the Vbench benchmark.

---

### Official Review · Reviewer_7YkR · 2025-07-06

**Clarity:** 4
**Significance:** 4
**Originality:** 4
**Rating:** 5
**Confidence:** 4

**Summary:**

This paper proposes Infinity*, a unified spacetime autoregressive framework for image and video synthesis under a purely discrete, autoregressive pipeline. Building on the prior Infinity and VAR frameworks, it extends next-scale prediction to videos by introducing a spacetime pyramid that incorporates the temporal axis. To enable this extension effectively, the paper introduces: (1) knowledge inheritance from continuous video VAEs for efficient discrete tokenizer training, (2) stochastic quantizer depth to address information imbalance across scales in the tokenizer, (3) semantic scale repetition to improve the generation quality of semantic tokens in early scales, and (4) spacetime sparse attention to reduce computational cost. The framework supports text-to-image, text-to-video, image-to-video, and video extrapolation within a single model, achieving state-of-the-art or near state-of-the-art results. It outperforms prior autoregressive methods and even some diffusion methods such as HunyuanVideo while being 10× faster than leading diffusion models when generating 5s 720p videos.

**Questions:**

Regarding semantic scale repetition, I would like clarification on whether the early scales are generated multiple times (each round conditioned on all preceding rounds) with supervision applied only to the final round. Additionally, I am interested in understanding how much quantitative improvement this repetition design brings to generation quality.

**Ethical Concerns:**

["NO or VERY MINOR ethics concerns only"]

**Final Justification:**

This work is a well-designed extension of VAR framework to video generation and achieves competitive performance. The authors conducted extensive experiments to address my concerns. I believe this work can benefit the community of AR video generation.

**Limitations:**

Yes

**Quality:**

4

**Strengths And Weaknesses:**

Strengths:

1. The paper proposes a highly effective framework that extends next-scale prediction to video generation, enabling support for text-to-image, text-to-video, image-to-video, and video extrapolation tasks within a single unified model.

2. The proposed framework achieves very strong performance, attaining state-of-the-art or near state-of-the-art results on benchmarks such as VBench and GenEval while significantly outperforming prior autoregressive methods and even surpassing some diffusion models like HunyuanVideo.

3. I enjoyed reading this paper; it is well written, with clear and well-motivated design choices for both the tokenizer and transformer components, supported by ablation studies that validate these decisions.

Weaknesses:

This is not strictly a weakness, but I am curious about the proposed Spacetime Sparse Attention design, which attends only to the last scale of the preceding clip. Since I assume the last scale contains the most tokens, I would like to understand how much computation this sparse attention actually saves and how much, if any, performance is lost compared to using full attention. I wonder why the current clip’s scales do not condition on the smaller scales of the preceding clip, which seems more natural and would likely save more computation. Has this alternative been tried, and how does it compare in terms of efficiency and quality?

---

> ### Author Rebuttal · Authors · 2025-07-31
>
> ## Response to Reviewer ``7YkR`` (Rating: Accept)
>
> We thank reviewers for your valuable comments on Infinity$^*$, acknowledging strong performance with fast speed (``7YkR``,
> ``qR4k``, ``vUvY``, ``tNTA``, ``KwMu``), well-motivated design (``7YkR``, ``vUvY``, ``KwMu``), valuable contribution to the generative model community (``qR4k``, ``tNTA``, ``KwMu``), with solid experimental results (``qR4k``, ``KwMu``). Your advice will help us enhance the paper.
>
> #### **Q1:  About the proposed spacetime sparse attention, how much computation does it save and how much performance loss does it incur compared to full attention?**
> In Table 1, we compare our spacetime sparse attention with full attention. Spacetime sparse attention shows superior performance to full attention in the Vbench total score (81.28 v.s. 80.77), while showing a significant advantage in saving computation and GPU VRAM. It's worth noting that spacetime sparse attention achieves 1.5$\times$ speedup when generating 192p 161 frames. The efficiency advantage becomes larger as the resolution and duration grow. For 480p 161 frames, full attention fails due to OOM while spacetime sparse attention completes it within 44.7s using 63GB VRAM. We conclude that spacetime sparse attention significantly improves efficiency without performance loss. We hypothesize that only attending to the previous clip's largest scale yields better results than full attention because it reduces exposure bias. In full attention, since the context is all ground truth during training, it is more susceptible to accumulated errors during inference.
>
>
>
> | Method                                    | total score | quality score | semantic score |
> |---|:---:|:---:|:---:|
> | Full Attention                                 | 80.77       | 81.15         | 79.23          |
> | Attend to former clip's largest scale     | **81.28**       | **81.56**         | **80.16**          |
> | Attend to former clip's 3rd largest scale | 80.86       | 81.26         | 79.26          |
> | Attend to former clip's 6th largest scale | 80.75       | 80.98         | 79.80          |
>
> Table 1. Performance comparison of various attention mechanisms
>
>
> #### **Q2:  Why the current clip do not condition on smaller scales of the preceding clip, which seems more natural and would likely save more computation. Has this alternative been tried, and how does it compare in terms of efficiency and quality?**
>
> The reason we do not condition on smaller scales of the preceding clip is that it misses the former clips' visual details and brings severe visual inconsistency between clips. As illustrated in Table 2, although conditioning on smaller scales  reaches 1.1$\times$, 1.5$\times$ speedup for 192p and 480p 161 frames, we observe a significant performance drop in Vbench from 81.28 to 80.75 as shown in Table 1. Our qualitative comparison also shows that attending to smaller scales bring obvious visual inconsistency between clips. Therefore, attending to the largest scale of the preceding clip strikes a better balance between computational efficiency and visual quality.
>
> |      Method                           | (192p 65 frames, 4s 16fps) | (192p 161 frames, 10s 16fps) | (480p 161 frames, 10s 16fps) |
> |---|:---:|:---:|:---:|
> | Full Attn                               | 8.6s / 40.8GB              | 24.3s / 57GB                 | OOM                          |
> | Attend to former clip's largest scale   | 7.7s / 38.5GB              | 16.7s / 40GB                 | 44.7s / 63GB                 |
> | Attend to former clip's 3rd largest scale | 7.4s / 38.2GB              | 15.8s / 39GB                 | 34.5s / 58GB                 |
> | Attend to former clip's 6th largest scale | 7.3s / 37.9GB              | 15.2s / 38GB                 | 30.5s / 55GB                 |
>
> Table 2. Computational efficiency comparison of attention mechanisms measured on a single NVIDIA H100 GPU with 8B parameters.
>
>
> #### **Q3:  Whether the early scales are generated multiple times with supervision applied only to the final round?**
> That's not the case. As a matter of fact, repeated early scales are treated as independent scales in VAR's residual quantification procedure. Therefore, both the original scale and the $n$ repeated scales receive their own corresponding ground truth and supervision signals.
>
> #### **Q4:  How much quantitative improvement does this repetition design bring to generation quality?**
> As shown in Table 3 , semantic scale repetition improves VBench's overall score from 75.72 to 80.97. More repetition literally yileds better results from 80.97 to 81.28. Here we set the repetition times $n=6$ to balance the performance and efficiency.
>
> | Method |  total score  |  quality score  |   semantic score   |
> |---|:---:|:---:|:---:|
> | w/o SSR | 75.72 | 76.73 | 71.68 |
> | With SSR (repetition=4) | 80.97 | 81.38 | 79.32 |
> | With SSR (repetition=6) | **81.28** | **81.56** | **80.16** |
>
> Table 3. Comparison of different repetition times in Semantic Scale Repetition (SSR)
>
>
> #### **Experiments Setup**
> All our ablation experiments were conducted using a 1M training set with a batch size of 40, trained at 192p resolution for 30,000 iterations. We evaluated the results on the Vbench benchmark.

---

> > ### Comment · Reviewer_7YkR · 2025-08-05
> >
> > Thank you for your feedback. The authors' response addresses my concerns.

---

> > > ### Author Response · Authors · 2025-08-05
> > >
> > > We are glad our response addressed your concerns, and we truly appreciate your thoughtful comments and support.

---

### Comment · Area_Chair_hMdK · 2025-08-03

Dear Reviewers,

Please:

- Read all reviews and author responses carefully;
- Join the discussion, especially if there are any remaining questions;
- Share your first comment as early as you can to allow time for a productive author-reviewer discussion.

Thank you for your responsible reviewing.

Best,

AC

---

### Decision · Program_Chairs · 2025-09-17

**Decision:**

Accept (oral)

**Comment:**

This paper extends the VAR framework to video generation by adding the temporal dimension into its pyramid design. The authors introduce a new sparse attention mechanism that conditions only on the final scale of the previous clip, striking a balance between latency and temporal consistency. They also propose additional techniques to improve generation quality, such as repeatedly generating the first few scales and training the tokenizer with stochastic quantizer depth.

The reviewers gave consistently positive feedback, highlighting the method’s versatility for video generation tasks, computational efficiency, strong empirical results, and thorough ablation studies. Some concerns were raised about design choices, latency analysis, and the need for more empirical evidence and implementation details to support the claims. The authors addressed these issues well, and it is strongly recommended that they include the corresponding clarifications and results in the revision.

Overall, this paper introduces a scalable visual autoregressive generation framework that can handle challenging long-duration video generation tasks, while also unifying text-to-image, text-to-video, and image-to-video generation in a single model. This is an important step forward in the visual generation field, complementing existing autoregressive and diffusion approaches.
Empirically, the paper demonstrates both strong performance and efficiency. It surpasses prior AR models and even some diffusion baselines, while generating videos about 10 times faster. This makes the work highly impactful for real-world use. In addition, the newly designed modules are insightful and could inspire further research.
Besides, the reviewers were consistently positive about its versatility, efficiency, and thoroughness.
Considering these points, the AC recommended this paper for an oral presentation.